# Gene–Diet Interactions: Viability of Lactoferrin-Fortified Yoghurt as an Element of Diet Therapy in Patients Predisposed to Overweight and Obesity

**DOI:** 10.3390/foods12152929

**Published:** 2023-08-02

**Authors:** Anna Jańczuk-Grabowska, Tomasz Czernecki, Aneta Brodziak

**Affiliations:** 1Department of Quality Assessment and Processing of Animal Products, Faculty of Animal Sciences and Bioeconomy, University of Life Sciences in Lublin, 20-950 Lublin, Poland; annajanczuk44@gmail.com; 2Department of Biotechnology, Microbiology and Human Nutrition, Faculty of Food Science and Biotechnology, University of Life Sciences in Lublin, 20-704 Lublin, Poland

**Keywords:** lactoferrin, bioactive potential, nutrigenomics, nutrigenetics, obesity, overweight, carbohydrate metabolism disorders, personalised nutrition, functional yoghurt

## Abstract

Given the availability of molecular tools, population studies increasingly include the gen-diet interactions in their considerations. The use of these interactions allows for the obtaining of more uniform research groups. In practice, this translates into the possibility of reducing the size of the research group while maintaining the precision of the research. The research results obtained in this way can be used to select certain ingredients and foods in a dietary intervention with a higher degree of personalisation. In both prophylaxis and dietary therapy of overweight and obesity, the proper selection of bioactive ingredients best suited to the given group of consumers is of key importance. Hence, the aim of the presented study was to assess the effectiveness of a dietary intervention with the use of lactoferrin (LF)-fortified yoghurt, in terms of the ability to regulate body weight and carbohydrate metabolism in individuals whose genomes contained single nucleotide polymorphisms that predisposed them to increased accumulation of fatty tissue and consequently overweight or obesity. A group of 137 participants (98 women and 37 men) of Polish origin were screened for the presence of four single nucleotide polymorphisms (rs993960—FTO gene, rs7903146—TCF7L2 gene, rs10830963—MTNR1B gene, and rs1121980—FTO gene). Subsequently, a group of 19 participants diagnosed with the presence of risk factors within said SNPs underwent a 21-day dietary intervention (crossover study) with the use of yoghurt fortified with lactoferrin (200 mg/day). The results of the study revealed a genetic difference between the Polish population and the European average, in terms of the SNPs analysed. The dietary intervention showed a statistically significantly higher efficiency in terms of body mass reduction (*p* = 0.000) and lowering the glycated haemoglobin ratio (HbA1c) (*p* = 0.000) when consuming specially prepared yoghurt containing lactoferrin, as compared to results registered for unfortified yoghurt. Given the above, yoghurt fortified with LF should be considered as a viable element of diet therapy in overweight and obese patients diagnosed with risk factors within the analysed polymorphisms.

## 1. Introduction

Currently, obesity has become one of the most common health problems in the world [1]. The incidence of obesity tripled in the period from 1976 to 2016. In 2016, it was estimated that as many as 13% of the adult population were obese and 39% of adults (over the age of 18) were overweight [2].

Obesity is a multi-factorial condition. Its emergence results from a number of factors including excessive calory intake and lack of movement, as well as genetic factors. Some polymorphisms in the FTO gene (fat mass- and obesity-associated) have been strongly associated with elevated BMI [3] and type 2 diabetes [4]. The gene is located in chromosome 16 and encodes the alpha-ketoglutarate-dependent dioxygenase enzyme [5]. It plays a major role in the accumulation of adipose tissue as the enzyme it encodes contributes to the mechanisms of thermogenesis, energy homeostasis, adipocyte differentiation, and metabolic rate [6,7]. The A allele single nucleotide polymorphism rs9939609 located in FTO gene is strongly associated with the accumulation of fatty tissue because it participates in the demethylation of N6-methyladenosine ghrelin. The process impacts the levels of ghrelin and acyl-ghrelin by inducing increased food consumption [8]. The risk of obesity in carriers of two risk alleles (AA) is elevated by between 20 and 30% [9]. It has been evidenced that single nucleotide polymorphism (SNP) rs1121980 in the FTO gene is also related to an increased risk of class II (BMI 35.0–39.9 kg/m^2^) and III (BMI ≥ 40.0 kg/m^2^) obesity in women in the Mexican population [10]. In the Korean population, individuals with BMI ≥ 25 who were carriers of the T allele were found to have a particular preference for consuming fatty foods, i.e., fish, coffee cream, candy, and snacks [11]. In the Slovak population, the CC variant of the polymorphism rs1121980 has a protective effect against obesity compared to the TT variant. The most common variant of the polymorphism in the Slovak population is the CT variant found in 40.5% of people, and the rarest CC variant is present in 29.1% of people [12].

Another single nucleotide polymorphism associated with overweight and obesity is rs7903146, located in TCF7L2 (gene encoding the TCF7L transcription factor). TCF7L2 influences the expression of the proglucagon gene regulating blood glucose homeostasis. The presence of the T allele is associated with an increased risk of non-alcoholic lipid infiltration, type 2 diabetes, glucose tolerance disorders, and hepatic β-cell disfunction and can also contribute to insulin resistance [13,14,15,16]. Body mass reduction usually improves glucose metabolism in obese children. However, in the 2008 study by Reinehr et al. [14], a negative effect in terms of insulin resistance and insulin sensitivity was observed in children with the T allele in polymorphism rs7903146.

Polymorphism of a single nucleotide (SNP), rs10830963, in the melatonin receptor 1 gene (MTNR1B) may be involved in body mass homeostasis. The G allele is associated with an increased risk of diabetes, gestational diabetes, and obesity [17,18]. Studies indicate that SNP rs10830963 is demonstrably linked to body weight loss as well as secondary insulin-level changes in persons receiving a hypocaloric diet [18,19]. Male participants with CG/GG genotypes in polymorphism rs10830963 showed lesser body weight loss as compared to participants with the CC genotype. In female carriers of the G allele, higher consumption of total protein and animal protein led to lower weight loss [19]. Similarly, in a 2020 study by De Luis et al. [16], individuals with a single G allele receiving a low-calory, high-protein diet showed worse overall improvement in terms of the analysed parameters, i.e., total cholesterol, LDL cholesterol, insulin, triglycerides, and HOMA IR, as compared to individuals with CC genotype in rs10830963.

The research results cited above confirm that individuals whose genotypes include risk alleles within the single nucleotide polymorphisms rs9939609, rs1121980, rs7903146, and rs10830963 are more susceptible to obesity and carbohydrate metabolism disorders due to specific metabolic pathway biomechanisms. For this reason, it is important to identify the bioactive substances whose inclusion in staple diet, already from a young age, will have confirmed preventive effects in terms of the emergence and exacerbation of negative changes in the organism. At the same time, the effectiveness of said bioactive components ought to be verified in a group of participants diagnosed with the relevant genetic factors to ensure that the bioactive ingredient will indeed be effective in the given group of patients. As follows from the preliminary studies conducted, lactoferrin may prove to be one such ingredient.

Lactoferrin (LF) is an iron-binding glycoprotein with anti-inflammatory, antibacterial, anticancer, and immunity-boosting properties [20]. Lactoferrin supplementation may also be helpful in facilitating the treatment of overweight and obesity. It has been evidenced that lactoferrin has a positive impact on the reduction of visceral fat and triglycerides [21]. Xiong et al. [1] demonstrated that the administration of 100 mg of bovine lactoferrin (bLF)/per day for 15 weeks to mice suffering from obesity induced with a high-fat diet facilitated a reduction in body weight gain and visceral obesity, as well as serum levels of lipids, leptin, and glucose. A similar model was analysed in 2016 by Sun et al. [20] who also administered 100 mg bLF to mice with high-fat diet-induced obesity for 12 weeks. They reported a reduction of inflammation, regulation of glucose levels, and modulation of the intestinal microbiota composition, including an increase in *Bifidobacterium* spp. counts and inhibition of *Enterobacteriales* growth [22]. In mice receiving water containing 2% of lactoferrin, researchers observed a significant decrease in visceral fat levels, total cholesterol, LDL cholesterol, triglycerides, and glucose. Additionally, certain effects in terms of intestinal microbiota regulation were also reported [20]. The cited reports corroborate the relevance of the undertaken research. As such, the aim of our study was to assess the effectiveness of supplementation with LF-fortified yoghurt in terms of weight regulation and carbohydrate metabolism improvement in participants whose genomes contained single nucleotide polymorphisms that predisposed them to excessive fatty tissue accumulation and consequently overweight or obesity.

## 2. Materials and Methods

### 2.1. The Studied Population

The studied population was composed of male and female participants from various regions of Poland. Genotyping screenings were conducted in 135 individuals, 98 women and 37 men. Participants qualified for the subsequent analyses were exclusively individuals with a risk allele in 3 of 4 polymorphisms measured, with preference for individuals with the highest number of risk alleles rs9939609 (AA/AT), rs7903146 (TT/CT), rs10830963 (CC/CG), and rs1121980 (AA/AG), who had a body mass index (BMI) above 25 kg/m^2^ or a weight to height ratio (WHtR) of 49 or more and who consented to taking part in the study. The study involving dietary intervention was designed in a crossover study design (AB:BA) in which 19 participants were included, including 11 women and 8 men. Group A (the control group) had a dietary intervention with placebo for 7 days, and group B (the study group with lactoferrin—LF) obtained a dietary intervention with LF for 21 days. Then, in group A, after 7 days, a dietary intervention with lactoferrin was introduced (the study group with LF), and it lasted for 21 days, whereas group B, after 21 days, had a dietary intervention with a placebo (the control group), and it lasted for 7 days. The intervention study lasted 21 days, and the control group with placebo lasted 7 days, after which time the groups were switched. There was no transition period between switching of the groups. The applied research arrangement allows for the verification of the effect of the dietary intervention with placebo, conducted as the first one, on the results obtained in the group with lactoferrin. The dietary intervention lasted 28 days.

The study was positively evaluated the Bioethics Committee at the Medical University of Lublin, Poland, and conducted in accordance with research approval no. KE-0254/276/2017 of 23 November 2017.

### 2.2. Dietary Intervention

The study participants received individually planned diets composed of 4 meals: breakfast, lunch, dinner, and supper. The calculations of basic metabolism were conducted using Mifflin’s formula. The comprehensive metabolism (CPM) was calculated by factoring in the physical activity level (PAL). The protein demand in grams was calculated by multiplying the proper body weight (pbw) calculated from Lorentz’s formula by 1.1g of protein/kg pbw. The fat demand in grams was calculated by dividing 25% of the CPM by 9. The carbohydrate supply in grams was calculated by deducting the caloric value of proteins and fats from the CPM and dividing the result by 4. Each study participant was instructed on how to follow the menu and was obliged to register in their daily consumption journals any departure from the assigned diet. Throughout the period of the dietetic intervention, the participants received yoghurt produced as part of the experiment, 250 g daily, unfortified (placebo, control group) or fortified with lactoferrin dosed at 80 mg per 100 g (experimental group).

### 2.3. Genotyping

The DNA material was collected from the patients by swabbing from the inside of the cheek. The collected material contained epithelial cells and white blood cells, which are a good source of DNA. DNA samples were isolated using a Gene MATRIX Swab-Extract DNA purification Kit from EurX Polska (Gdańsk, Poland), in accordance with the procedure recommended by the manufacturer. The concentration and purity of the isolated DNA were evaluated with a DS-11 spectrophotometer from DeNovix (Wilmington, DE, USA), based on the absorption levels at the wavelengths of 260 nm and 280 nm. The quality of the isolated DNA was determined by way of electrophoresis in aerated gel (1%) and UV visualisation with a SimplySafe stain from EurX Polska (Gdańsk, Poland). The remaining portion of the DNA isolate was labelled and stored at −20 °C for further study.

The genotyping focused on four polymorphisms of a single nucleotide: rs9939609, rs7903146, rs10830963, and rs1121980. The polymerase chain reaction (PCR) was done with the use of a Biorad T-100 thermocycler (BioRad, CA, USA). The primer of the PCR reaction was designed with the use of Primer 3 software ver. 4.1.0 for each individual single nucleotide polymorphism. A total of 45 amplification cycles were performed. The amplification of the desired product was confirmed with electrophoresis in aerated gel (1.5%) and product visualisation in UV light using a SimplySafe stain from EurX Polska (Gdańsk, Poland).

Sanger sequencing of the obtained amplicons was conducted with a 3730 Genetic Analyzer capillary sequencer from Applied Biosystems (Waltham, MA, USA), using BigDye Terminator v3.1 Cycle Sequencing Kit reagents from Applied Biosystems (Waltham, MA, USA).

Blind testing as well as positive and negative controls were used in the laboratory. The result was read independently by 2 people and entered into the laboratory system. If both readings were identical, the system approved the analysis result.

### 2.4. Yoghurt Fortified with Lactoferrin

The yoghurt was produced form cow milk at one of the small dairy processing plants operating in the Lublin Voivodeship. The exact production process was described by Jańczuk et al. [23]. The yoghurt was distributed to study participants immediately directly from production, using adequate cold-chain methods.

### 2.5. Anthropometric Measurements

Measurements of body circumference were conducted in the upright position using a Gulick anthropometric tape with 0.1 cm accuracy. During the measurements, the tape remained parallel to the floor, and the participants were instructed to stand with their feet spread out to chest width. Waist measurements were taken at the middle point between the lower edge of the costal arch and the top of the iliac crest, during a regular exhalation. The hip circumference was measured at the widest point and the thigh (maximum circumference) below the gluteal fold, in a balanced posture. Each measurement was repeated twice. If the results varied by more than 0.7 cm, a third measurement was taken.

The participants’ height was measured in the upright position, standing straight with feet placed together, shoeless, from the floor to the anatomical top of the head—the vertex—making sure that the ear channel remained aligned with the cheek bone. The measurements were taken with an ADE MZ10038 stadiometer (Hamburg, Germany).

The measurements of body mass and adipose tissue were conducted with a Tanita MC 780 MA body composition monitor. Patients were weighed wearing only light clothes, shoeless, with mass distributed evenly across both feet.

Body mass index (BMI) was calculated as body weight in kg divided by the square of the body height in meters (kg/m^2^). The following interpretation criteria were adopted: 18.50–24.99—correct weight, 25.00–29.90—overweight, 30.00–34.99—1st degree obesity, and 35.00–39.99—2nd degree obesity.

The WtHR (waist-to-height ratio) was calculated as the ratio of waist circumference to height (both in cm) multiplied by 100. The following interpretation criteria were adopted: to 35.00 malnutrition, 35.01–42.00—underweight, 42.01–46.00—slight underweight, 46.01–54.00—overweight, 54.01–58.00—severe overweight, and above 58.00—obesity.

### 2.6. Statistical Analysis

The statistical analysis was performed with the use of Statistica ver. 13.1 (StatSoft Inc., Dell, Round Rock, TX, USA). The normal distribution of the results was verified with the Shapiro–Wilk test. As the test indicated lack of normal distribution of the results, which may have been due to the uniformity of the group resulting from the restrictive inclusion criteria, the statistical analysis was conducted using nonparametric tests, e.g., chi-squared, Wilcoxon, and Kruskal–Wallis tests and regression analyses. Results within the confidence interval of 95.0% at *p* ≤ 0.05 were considered statistically significant.

## 3. Results

### 3.1. Screening Genotyping

Figure 1, Figure 2, Figure 3 and Figure 4 present the results of the genotype screening conducted in the group of 135 participants from Poland, 98 women and 37 men. The data were compared to the distribution of polymorphisms in the European population using the 1000 Genomes database [24]. Figure 1 presents the distribution for SNP rs9939609. In our study, AA homozygotes represented as much as 32.6% of the population, while in the 1000 Genomes database the same was only 19.9%. The results in terms of AT heterozygotes were roughly consistent with the reference data—41.5% in the studied population compared to 42.9% in the database. TT homozygotes not carrying the risk variant represented only 25.9% in our study, as compared to 37.2% in the 1000 Genomes database [24].

Figure 2 presents the distribution of SNP rs1121980 variants. Differences in terms of the distribution of the highest-risk variant AA were clearly evident: in our study as many as 35.6% of the participants carried the variant while the European average was only 23.3%. The result for AG heterozygotes was 42.2% in our study, as compared to 42.1% in the reference database. Differences were also evidenced in terms of the GG variant distribution which was calculated at 22.2% in the present study, as compared to 34.6% in the 1000 Genomes database [24].

Figure 3 presents the distribution of genetic variants in SNP rs7903146. The most common variant by far, both in our study and the 1000 Genomes database [24], present in 53.3% and 48.1% of the participants, respectively, was the CC variant associated with carbohydrate metabolism disorders. The second most common was the CT heterozygotic variant, identified in 37.8% participants in our study, as compared to 40.4% in the 1000 Genomes database [24]. TT heterozygotes carrying two risk alleles were found in only 8.9% cases in our study relative to 11.5% in the reference data.

The distribution for the last SNP rs9939609, analysed in the screening test, is presented on Figure 4. The CC variant was slightly more common in the analysed population (51.1%) than in the 1000 Genomes database (50.3%) [24]. Greater differences in the distribution of genetic variants could be observed for the CG heterozygotic variant, present in 34.1% of the cases in our study and 41.7% of the cases in the reference data. The highest-risk factor, GG, was the least common in both datasets, corresponding to 14.8% on our study and 8.0% in the 1000 Genomes data [24].

### 3.2. Phenotypic Characteristics of the Study Participants

The study was conducted in a group of 19 participants, 11 women and 8 men. Table 1 presents the initial parameters measured in the study population. The mean age of the women was 35 years and of the men was 38.8 years. The mean body weight was statistically significantly (*p* = 0.009) lower in women as compared to men (88.7 kg) and averaged at 79.8 kg. The mean waist circumference was 96.3 cm in women and 101.8 cm in men (*p* = 0.032). The men and the women taking part in the study had very similar hip circumference parameters, respectively, 107.3 cm for women and 107.3 cm for men, as well as a nearly identical WHtR: women—58.5 and men—58.7, initial HbA1c value: women—5.8 and men—5.8, and BMI: women—29.4 kg/m^2^ and men—29.5 kg/m^2^. The mean thigh circumference was higher in women 59.1 cm as compared to men—58.4 cm. A significant (*p* = 0.001) difference between the two groups was observed in terms of fatty tissue %, which was 37.3% in women and 28.5% in men.

### 3.3. Genotypic Characteristics of the Study Participants

Table 2 presents a breakdown of genetic variants identified in the research participants. Four polymorphisms of a single nucleotide closely related to overweight and obesity, rs9939609, rs7903146, rs10830963, and rs1121980, were selected for the analysis. All the participants were characterised by the presence of at least one risk-factor allele—A in rs9939609, which has been linked to the risk of obesity, elevated BMI, and type II diabetes. Homozygotes AA constituted 84.21%, and heterozygotes AT constituted 15.79%. Allele A present in rs1121980 is a risk-factor allele linked to obesity and increased consumption of high-fat foods. All members of the study group showed the presence of at least a single risk-factor allele A, with 68.42% of the same carrying the AA variant and 31.51% carrying the AG variant. In the case of rs7903146, a polymorphism located in the TCF7L2 gene associated with blood glucose homeostasis, as many as 78.95% of the participants were carriers of the high-risk TT variant, while 21.05% carried the CT variant. The lowest percentage of the research participants carried at least a single G risk allele in rs10830963 linked to type I diabetes and obesity—42.11%—with GG homozygotes identified only in 10.53% and CG heterozygotes identified in 31.58%.

### 3.4. Effect of Consumption of Yoghurt with Lactoferrin on Body Weight

Figure 5 presents the body weight loss in the study participants. After 21 days, a statistically significant improvement in body mass parameters was observed. Under the 7d/21d regimen, the body weight decreased by 1.26 kg in men and 1.15 kg in women, as compared to only 0.06 kg in men and 0.07 kg in women in the control group. The size of the body weight reduction between men and women was statistically insignificant (*p* > 0.05). The size of weight reduction observed between the control group and the study group with LF was significant at the level *p* = 0.000. Under the 21d/7d regimen, the study group revealed very similar results in terms of body weight reduction to those registered for the 7d/21d regimen, men—1.07 kg and women—1.10 kg. The corresponding weight loss observed in the control group was 0.18 kg in men and 0.07 kg in women. No statistically significant (*p* > 0.05) differences were observed between the 21d/7d and 7d/21d experimental regimens. This suggests that the consumption of lactoferrin-free yoghurt had no impact on the participants’ body weight.

### 3.5. Effect of Consumption of Yoghurt with Lactoferrin on the Concentration of Glycated Haemoglobin (HbA1c)

Figure 6 illustrates the % changes of HbA1c levels. Under the 21/7 experimental regimen, a reduction of the HbA1c levels was observed—0.17% in men and 0.18% in women. The corresponding values for the control group were 0.02% in men and 0.01% in women. Similar to the results in terms of body mass reduction, the differences in % HbA1c reduction between the 7/21 and 21/7 experimental regimens were negligible and not statistically significant. The % HbA1c reduction observed was 0.21% in men and 0.17% in women. The corresponding value in the control group was 0.01% in both male and female participants.

## 4. Discussion

The results of the analysis indicate that consumption of yoghurt containing 80 mg/100 g bLF dosed at 250 g daily, in a population characterised by the presence of polymorphisms conducive to obesity and carbohydrate metabolism disorders, may facilitate weight loss and better regulation of the HbA1c level. Over the course of the 3-week experiment, the study participants lost more weight than the control group of participants, specifically 1.03 and 0.89 kg more for women and men, respectively, under the 21/7 regimen, and 1.08 kg and 1.20 kg more for women and men, respectively, under the 7/21 regimen.

The obtained results in terms of body mass reduction are consistent with those reported by other researchers. Hassan et al. [23] evidenced, in an animal model, that the consumption for 45 days of yoghurt containing LF, dosed at 50 and 100 mg per kg of body mass, reduced weight gain in mice. Similarly, in a study by Ono et al. [24] conducted on a human model, it was demonstrated that the consumption of bLF may reduce obesity in overweight participants without reducing the calorie intake. The effects of LF supplementation in terms of body mass reduction have also been corroborated in other scientific reports, but unfortunately, at the moment there are few studies in which the vehicle for LF was yoghurt. In most studies, LF was administered as an aqueous solution or oral supplementation [1,20,21]. For example, in a study by Li and Ma [18], mice on a high-fat diet were watered with drinking water containing 2% LF for 12 weeks. After this period, a reduction in visceral fat levels was noted with *p* < 0.05. In a study by Xiong et al. [1], mice fed a high-fat diet receiving 100 mg bLF/kg b.w. for 15 weeks showed significantly lower weight gain and visceral fat reduction. The positive effects of LF supplementation on fat reduction were also seen in the shorter supplementation period of 4 weeks, which is only one week longer than in our study. Morishita et al. [19] confirmed that administration of 100 mg of bLF to mice for a period of 4 weeks reduces mesenteric adipose tissue *p* < 0.05.

The weight loss results observed in the experimental group may have been influenced by factors other than the LF supplementation itself or the form thereof. Yoghurt is known to have certain unique properties that may be beneficial to health and conducive to reducing calorie intake and regulating glycaemia [25]. The results reported by Madjd et al. [26] suggest that introducing low-fat yoghurt to the diet at 400 g per day for 12 weeks in overweight and obese women of the I (BMI 30.0–34.9 kg/m^2^) and II (BMI 35.0–39.9 kg/m^2^) degrees may trigger weight loss of up to 5.03 kg. In addition, a 4.8 cm decrease in waist circumference was noted.

The HbA1c levels registered for the own study participants decreased by between 0.18 and 0.21%, depending on the group experimental regimen, under the 21/7 experimental regimen 0.17% in men and 0.18% in women, and for the 7d/21d regimen 0.21% men and 0.17% women. The positive effect of consuming yoghurt fortified with LF on carbohydrate metabolism was confirmed by Hassan et al. [23], although the specific value measured in that study was not HbA1c level but glucose concentration. Rats receiving 45 days of yoghurt fortified with 50 mg or 100 mg of LF per kg of body mass showed better results in terms of glucose level reduction as compared to the control group receiving a high-fat diet [26]. Similarly, in a study by Sun et al. [20], the administration of 100 mg/kg body mass bLF for 12 weeks significantly reduced blood glucose levels compared to mice that were only on a high-fat diet. Glucose levels were reduced to control values [22]. Other reports in which LF was not administered using yoghurt also noted positive effects on carbohydrate metabolism; e.g., in a study by Li and Ma [18], a 12-week watering of mice on a high-fat diet with 2% LF water resulted in a decrease in blood glucose levels with *p* < 0.05 [20]. In a study by Xiong et al. [1], feeding mice 100 mg of bLF for 15 weeks also reduced blood glucose levels.

The mechanism by which lactoferrin affects weight reduction and HBA1C remains unclear. However, there are scientific reports that describe the effect of lactoferrin on the weakening of adipogenesis in the context of modulation of (p172Thr) AMPK phosphorylation [27]. LF may also promote lipolysis by modulating the cAMP pathway and reducing perillipin expression [28].

The reduction of the HbA1c level after consuming the lactoferrin-fortified yoghurt for 3 weeks may also be due to the medium—yoghurt itself. As follows from the results reported by Madjd et al. [26], the consumption of low-fat yoghurt over a period of 12 weeks reduced the HbA1c level by 0.28% on average. The authors also observed benefits in terms of other parameters associated with carbohydrate metabolism, i.e., a reduction in plasma glucose by 0.24 mmol/L, insulin by 1.32 mU/mL, and HOMA-IR by 0.42.

One should bear in mind that the weight loss and HbA1c reduction results obtained in the present study may be less spectacular than those reported by other researchers cited above due to the specific genetic profile of the research participants, a much shorter study period, and the use of an animal model in the reports cited. Each participant was characterised by the presence of at least one risk allele in three of four SNP measurements (rs9939609, rs7903146, rs10830963, and rs1121980), which indicates a particular predisposition to obesity and carbohydrate metabolism disorders. The A allele at SNP rs9939609 contributes to increased food intake through its effect on the regulation of ghrelin and acyl ghrelin levels [5,9]. It has been proven that homozygotes at SNP rs9939609 have an increased risk of obesity by as much as 20–30% [9]. Carriers of at least one A allele at SNP rs1121980 are more prone to the occurrence of grade II (BMI 35.0–39.9 kg/m^2^) and III (BMI ≥ 40.0 kg/m^2^) obesity. Additionally, they are more likely to consume high-fat foods that contribute to excessive energy supply from the diet [10,11,12]. Also, the presence of the T allele at SNP rs7903146 is associated with the occurrence of carbohydrate disorders, i.e., lack of improvement in insulin sensitivity, and the G allele at SNP rs10830963 is associated with worse results in weight reduction while on a reduction diet [16,19]. Analysing the above reports, it seems reasonable to assume that the genetic profile presented by the study participants influenced not-so-high results in weight reduction and % HbA1c. However, it should be emphasised that the studies cited were mainly carried out with animals, i.e., mice and rats.

## 5. Conclusions

Due to individual metabolic differences resulting from, e.g., the presence of single nucleotide polymorphisms, the evaluation of nutrient bioactivity ought to be undertaken in as well-characterised consumer group as possible. Parameters such as height, age, or sex are not in themselves sufficient to fully determine metabolic characteristics, particularly in a situation of a prophylactic dietary intervention, i.e., in a case where no apparent symptoms of disease are present. The results of the conducted study indicate a higher incidence of certain SNP polymorphisms in the Polish population as compared to the European average. The conducted 21-day dietary intervention using lactoferrin-fortified yoghurt administered to individuals with a specific genetic profile with regard to selected SNPs revealed that consumption of the product facilitated reduction of body mass and the levels of glycated haemoglobin (HbA1c). The obtained results suggest that the product may be used as a viable therapeutic component in diet therapy. It can be concluded that the consumption of LF-fortified yoghurt may also prove to have prophylactic benefits in individuals with specific genetic predispositions. However, the assessment of the exact prophylactic viability of specific foods requires further in-depth study, including in terms of establishing a uniform protocol for the evaluation of the prophylactic benefits of a given component or food product. The study showed the therapeutic effect of lactoferrin use in a group of people with excessive body weight with specific genetic predispositions resulting from the possessed risk alleles of single nucleotide polymorphisms (rs9939609, rs1121980, rs7903146, and rs10830963). For this reason, the therapeutic effectiveness of lactoferrin should be confirmed on groups with other genetic variants. In addition, in the future, the interactions and phenotypic effect that may result from other SNP polymorphisms not included in this study should be determined.

## Figures and Tables

**Figure 1 foods-12-02929-f001:**
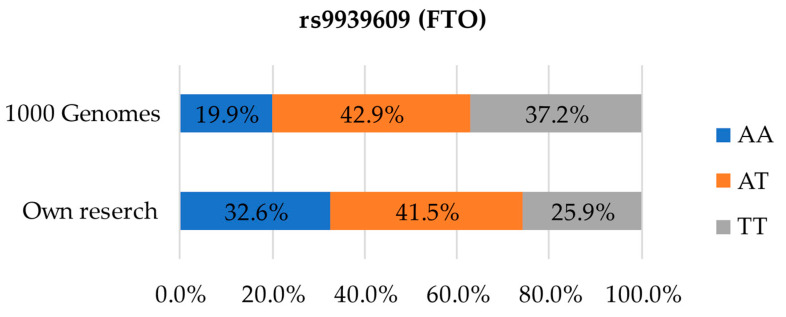
Distribution of genetic variants in SNP rs9939609 in the studied population compared to the European population.

**Figure 2 foods-12-02929-f002:**
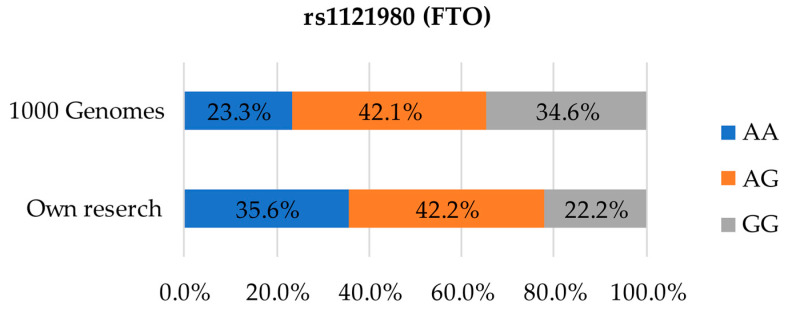
Distribution of genetic variants in SNP rs1121980 in the studied population compared to the European population.

**Figure 3 foods-12-02929-f003:**
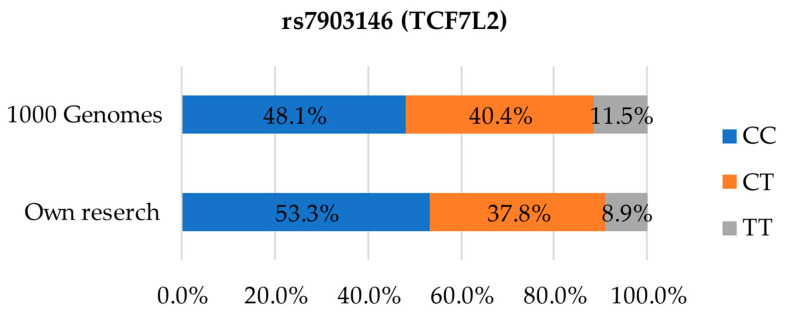
Distribution of genetic variants in SNP rs7903146 in the studied population compared to the European population.

**Figure 4 foods-12-02929-f004:**
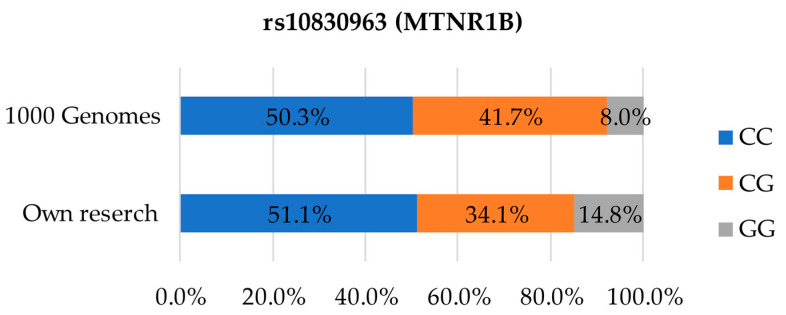
Distribution of genetic variants in SNP rs10830963 in the studied population compared to the European population.

**Figure 5 foods-12-02929-f005:**
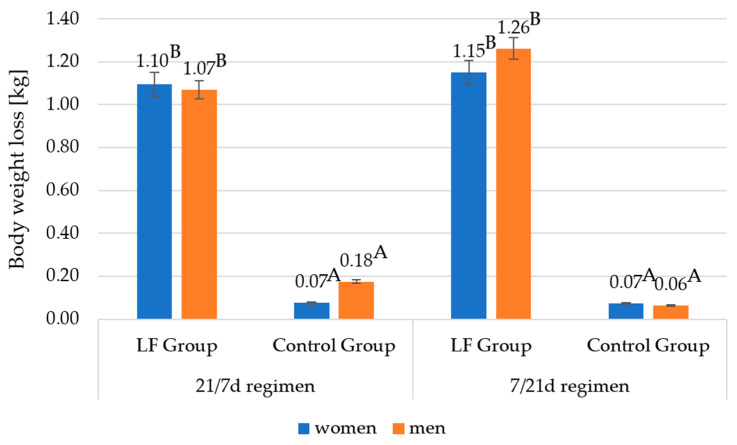
Body weight loss of study participants in the 21/7 day and 7/21 day regimen. A, B —differences between the research groups within the sex; A, B—significant differences at *p* ≤ 0.01.

**Figure 6 foods-12-02929-f006:**
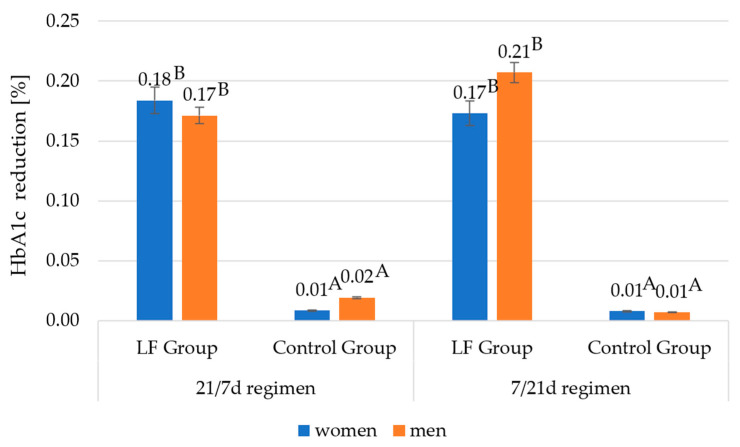
Reduction of the concentration of HbA1c of study participants in the 21/7 day and 7/21 day regimen. A, B—differences between the research groups within the sex; A, B—significant differences at *p* ≤ 0.01.

**Table 1 foods-12-02929-t001:** Characteristics of the study participants (x ± SD).

Parameters	Women	Men	*p*-Value
Age (years)	35.0 ± 5.2	38.8 ± 7.3	0.536
Height (cm)	164.0 ± 3.9	173.3 ± 4.0	0.001
Weight (kg)	79.8 ± 5.4	88.7 ± 6.8	0.009
Waist (cm)	96.3 ± 3.9	101.8 ± 4.5	0.032
Hips (cm)	107.3 ± 4.8	107.3 ± 3.6	0.967
Thigh (cm)	59.1 ± 2.6	58.4 ± 1.5	0.563
Body fat content (%)	37.3 ± 2.0	28.5 ± 2.3	0.001
BMI (kg/m^2^)	29.4 ±1.5	29.5 ± 1.5	0.901
WHtR	58.5 ± 2.3	58.7 ± 2.2	0.869
HbA1c (%)	5.8 ± 0.4	5.8 ± 4.4	0.710

**Table 2 foods-12-02929-t002:** Distribution of the risk variant predisposing to obesity and carbohydrate metabolism disorders in the population (%).

Gene	SNP	Genetic Variant *	Women	Men	Population
FTO	rs9939609	AA	100.00	62.50	84.21
		AT	0.00	37.50	15.79
		TT	0.00	0.00	0.00
FTO	rs1121980	AA	81.82	50.00	68.42
		AG	18.18	50.00	31.58
		GG	0.00	0.00	0.00
TCF7LN	rs7903146	TT	72.73	87.50	78.95
		CT	27.27	12.50	21.05
		CC	0.00	0.00	0.00
MTNR1B	rs10830963	GG	0.00	25.00	10.53
		CG	18.18	50.00	31.58
		CC	81.82	25.00	57.89

* rs9939609—A (risk allele), rs1121980—A (risk allele), rs7903146—T (risk allele), rs10830963—G (risk allele).

## Data Availability

Data is contained within the article.

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
