# Peer review of "Gene–Diet Interactions: Viability of Lactoferrin-Fortified Yoghurt as an Element of Diet Therapy in Patients Predisposed to Overweight and Obesity"

_foods, 2023, doi:10.3390/foods12152929_

Round 1

Reviewer 1 Report

This is interesting work that would appeal to those involved in the field of nutrigenetics. However, I believe that the manuscript needs more work before it would be publishable. Including editing for spelling and grammar, I have concerns about the following.

Page 2 Lines 46 and 52. These references are not appropriate. Original sources should be cited.

Lines 54-5 and lines 340, 374. What is Class I, II and III obesity? This should be explained.

Line 76 Authors report gender effect in reference 16 however there is no reported gender effect in this particular study. I also query use of the term gender to describe (biological) sex.

Page 3 line 120. What do authors mean by homozygotic systems? The study focuses on risk alleles, not homozygous genotypes; some are heterozygous eg. AT of rs9939609 This should be amended to accurately describe how the intervention focused on individuals with certain risk alleles.

Line 132. What is elevenses? Is this lunch? If so, the term lunch should be used.

Line 146. Red blood cells are not a source of DNA! This needs to be amended.

Questions about the study design:

Why did placebo last for 7 days and not 21 days like the intervention? The 21/7 regimen is not well explained.

What were the control methods used for genotyping? There should be controls and this information should be declared.

The manuscript needs editing as there are numerous spelling and grammar errors.

Author Response

Dear Reviewer 1,

Thank you for giving us the opportunity to submit a revised version of our manuscript titled „ Gene-Diet Interactions: Viability of Lactoferrin-Fortified Yoghurt as an Element of Diet Therapy in Patients Predisposed to Overweight and Obesity” to Foods.

We appreciate the time and effort that you have dedicated to providing your valuable feedback on our manuscript. We are grateful for your comments on our paper. We have made corrections in the text in accordance with the suggestions.

Reviewer: Page 2 Lines 46 and 52. These references are not appropriate. Original sources should be cited.

Authors: Thank you for the comment. This has been corrected and the text has been modified (Lines 47, 51-54).

Reviewer: Lines 54-5 and lines 340, 374. What is Class I, II and III obesity? This should be explained.

Authors: Thank you for your comment. The explanation has been added (Lines 57, 363, 404).

Reviewer: Line 76 Authors report gender effect in reference 16 however there is no reported gender effect in this particular study. I also query use of the term gender to describe (biological) sex.

Authors: Thank you for your attention. The information about the gender effect has been removed (Line 76).

Reviewer: Page 3 line 120. What do authors mean by homozygotic systems? The study focuses on risk alleles, not homozygous genotypes; some are heterozygous eg. AT of rs9939609 This should be amended to accurately describe how the intervention focused on individuals with certain risk alleles.

Authors: Preference was given to individuals with the highest number of risk alleles, i.e., containing polymorphism in both genes (risk homozygous) (Lines 122-123).

Reviewer: Line 132. What is elevenses? Is this lunch? If so, the term lunch should be used.

Authors: Thank you for your comment. Elevenses has been corrected for lunch (Line 142).

Reviewer: Line 146. Red blood cells are not a source of DNA! This needs to be amended.

Authors: Thank you for your attention - translation error. Red blood cells has been corrected for white blood cells in the paper (Line 156).

Questions about the study design:

Reviewer: Why did placebo last for 7 days and not 21 days like the intervention? The 21/7 regimen is not well explained.

Authors: Our other research have shown that 7 days is enough. This is also confirmed by comparing the results for the placebo group in the 7/21 and 21/7 days system. The volunteers were divided into 2 groups, group A and group B. Group A (the control group) had a dietary intervention with placebo for 7 days, and group B (the study group with lactoferrin – LF) obtained a dietary intervention with LF for 21 days. Then, in the group A, after 7 days, a dietary intervention with lactoferrin was introduced (the study group with LF) and it lasted for 21 days. Whereas the group B, after 21 days, had a dietary intervention with a placebo (the control group) and it lasted for 7 days. Such a research setup allows for estimating the effect of the first placebo dietary intervention on the results obtained in the group with lactoferrin. In studies without AB:BA rotation, the obtained result may be burdened with an error resulting from the effect of the placebo group, e.g. regulating meals, and this error may to some extent increase with the extension of the experiment time in the placebo group. On the other hand, running the placebo group and the study group in parallel poses a risk that unbalanced groups in terms of behaviour, including dietary behaviour, which may affect the results obtained (Lines 127-137).

Reviewer: What were the control methods used for genotyping? There should be controls and this information should be declared.

Authors: Blind testing as well as positive and negative controls were used in the laboratory. The result was read independently by 2 people and entered into the laboratory system. If both readings were identical, the system approved the analysis result (Lines 175-177).

Reviewer: Comments on the Quality of English Language. The manuscript needs editing as there are numerous spelling and grammar errors.

Authors: The manuscript has been linguistically corrected.

Kind regards,

Authors

Reviewer 2 Report

Very interesting study investigate the effects of lactoferrin-fortified yoghurt on weight loss among carriers of risk alleles for obesity, but there is some area need to be address

Abstract

This sentence need rephrase it is not clear and reword "This approach allows one to make study populations more uniform, hence potentially increasing the strength of the results and facilitating better selection of ingredients and foods to be included in dietary interventions."

The 4 SNPs chosen in the study located at which gene or genes

Add p value for these results in the abstract "The dietary intervention showed a higher efficiency in terms of body mass reduction and lowering the glycated haemoglobin ratio (HbA1c) when consuming specially prepared yoghurt containing lactoferrin, as compared to results registered for unfortified yoghurt."

The abbreviation "LF" introduce for first time in line 31 but should first mention in line 19.

The introduction is comprehensive and well written.

Methods

Why is the period of intervention group (LF) lasted 21 days while the control group with placebo lasted 7 days? This need justification.

Is there a wash up period between two arms? This need to be mentioned clearly.

Is elevenses the lunch meal? If so write lunch because it is widely popular.

Statistical analysis

Given this is genetic study, it is really important to adjust for covariates such as age gender, physical activity. I recommend running analysis using Univariate regression given this test is suitable skewed data. Authors can use log 10 for skewed data so they can use it in Univariate regression.

Result

Add p value for difference between women and men in table 1 and adjust paragraph in the section Characteristics of the study participants according to that.

Body weight

Add p value for statical significant results.

Please mention whether reduction in the result is statistically significant by adding P value.

Discussion

How yogurt and lactoferrin affect body weight and HBA1C? possible mechanism needs to be mentioned

Strengths and limitations of study need to be clearly stated.

Author Response

Dear Reviewer 2,

Thank you for giving us the opportunity to submit a revised version of our manuscript titled „ Gene-Diet Interactions: Viability of Lactoferrin-Fortified Yoghurt as an Element of Diet Therapy in Patients Predisposed to Overweight and Obesity” to Foods.

We appreciate the time and effort that you have dedicated to providing your valuable feedback on our manuscript. We are grateful for your comments on our paper. We have made corrections in the text in accordance with the suggestions.

Reviewer: Abstract This sentence need rephrase it is not clear and reword "This approach allows one to make study populations more uniform, hence potentially increasing the strength of the results and facilitating better selection of ingredients and foods to be included in dietary interventions."

Authors: Thank you for your comment. The sentence has been corrected (Lines 14-18).

Reviewer: The 4 SNPs chosen in the study located at which gene or genes

Authors: Thank you for your comment. The genes in which the tested SNPs are located have been added to the article (Lines 25-26).

Add p value for these results in the abstract "The dietary intervention showed a higher efficiency in terms of body mass reduction and lowering the glycated haemoglobin ratio (HbA1c) when consuming specially prepared yoghurt containing lactoferrin, as compared to results registered for unfortified yoghurt."

Authors: Thank you for your comment. The p-value has been added (Lines 30-32).

Reviewer: The abbreviation "LF" introduce for first time in line 31 but should first mention in line 19.

Authors: Thank you for your comment. The abbreviation "LF" has been introduced (Line 21).

Reviewer: Methods Why is the period of intervention group (LF) lasted 21 days while the control group with placebo lasted 7 days? This need justification.

Authors: Our other research have shown that 7 days is enough. This is also confirmed by comparing the results for the placebo group in the 7/21 and 21/7 days system. The volunteers were divided into 2 groups, group A and group B. Group A (the control group) had a dietary intervention with placebo for 7 days, and group B (the study group with lactoferrin – LF) obtained a dietary intervention with LF for 21 days. Then, in the group A, after 7 days, a dietary intervention with lactoferrin was introduced (the study group with LF) and it lasted for 21 days. Whereas the group B, after 21 days, had a dietary intervention with a placebo (the control group) and it lasted for 7 days. Such a research setup allows for estimating the effect of the first placebo dietary intervention on the results obtained in the group with lactoferrin. In studies without AB:BA rotation, the obtained result may be burdened with an error resulting from the effect of the placebo group, e.g. regulating meals, and this error may to some extent increase with the extension of the experiment time in the placebo group. On the other hand, running the placebo group and the study group in parallel poses a risk that unbalanced groups in terms of behaviour, including dietary behaviour, which may affect the results obtained (Lines 127-137).

Reviewer: Is there a wash up period between two arms? This need to be mentioned clearly.

Authors: There was no transition period between switching of the groups. Extending the study period brings more interfering factors, and the method of conducting the study (AB:BA test) protects against errors resulting from the time and order of placebo introduction (Lines 134-135).

Reviewer: Is elevenses the lunch meal? If so write lunch because it is widely popular.  

Authors: Thank you for your comment. Elevenses has been corrected for lunch (Line 142). 

Reviewer: Statistical analysis. Given this is genetic study, it is really important to adjust for covariates such as age gender, physical activity. I recommend running analysis using Univariate regression given this test is suitable skewed data. Authors can use log 10 for skewed data so they can use it in Univariate regression.

Authors: Thank you for your comment. Authors conducted the indicated statistical analysis (Lines 213-214). The result of the statistical analysis showed that age and gender had no statistically significant effect on changes in body weight and glycated hemoglobin.

Reviewer: Result Add p value for difference between women and men in table 1 and adjust paragraph in the section Characteristics of the study participants according to that.

Authors: Thank you for the comment. This has been added and the text has been modified (Lines 262-274, Table 1).

Reviewer: Body weight  Add p value for statical significant results. Please mention whether reduction in the result is statistically significant by adding P value.

Authors: Thank you for the comment. This has been added and the text has been modified (Lines 299-329, Figures 5 and 6).

Reviewer: Discussion  How yogurt and lactoferrin affect body weight and HBA1C? possible mechanism needs to be mentioned

Authors: Thank you for your advice. Possible mechanism has been added in the Discussion (Lines 380-384).

Reviewer: Strengths and limitations of study need to be clearly stated.

Authors: Thank you for the comment. This has been added and the text has been modified (Lines 430-436).

Kind regards,

Authors